# The Prevalence of “at Risk” Eating Disorders among Athletes in Jordan

**DOI:** 10.3390/sports10110182

**Published:** 2022-11-18

**Authors:** Hadeel A. Ghazzawi, Omar A. Alhaj, Lana S. Nemer, Adam T. Amawi, Khaled Trabelsi, Haitham A. Jahrami

**Affiliations:** 1Department Nutrition and Food Technology, School of Agriculture, The University of Jordan, Amman 11942, Jordan; 2Department of Nutrition, Faculty of Pharmacy and Medical Sciences, University of Petra, Amman 1196, Jordan; 3Department of Physical and Health Education, Faculty of Educational Sciences, Al-Ahliyya Amman University, Al-Salt 19328, Jordan; 4High Institute of Sport and Physical Education of Sfax, University of Sfax, Sfax 3000, Tunisia; 5Research Laboratory Education, Motricity, Sport and Health, EM2S, LR19JS01, University of Sfax, Sfax 3000, Tunisia; 6Ministry of Health, Manama 410, Bahrain; 7Department of Psychiatry, College of Medicine and Medical Sciences, Arabian Gulf University, Manama 323, Bahrain

**Keywords:** eating disorders, athletes, Jordan, sport type, EAT-26 items

## Abstract

Eating disorders (EDs) are addressed as one of the expanding mental health problems worldwide. While an ED is a clinical psychiatric diagnosis that can only be established after a psychiatric assessment, it is important to note that “at-risk” refers to people who will exhibit aberrant eating patterns but do not fully meet the requirements for an ED diagnosis. This study was designed to address the ED symptoms (i.e., “at-risk”) in Jordanian athletes and their association with age, sex, body mass index (BMI), and type of sport. A convenient, cross-sectional study was conducted among 249 athlete participants by answering the Eating Attitude Test (EAT-26). The EAT-26 results indicated an ED prevalence of 34% among Jordanian athletes. Within “at-risk” ED athletes, sex, age, and BMI had no significant differences in the rates of EDs. Outdoor sports had the least effect on EDs, while the highest was amongst gymnastics. EDs prevalence is alarming among Jordanian athletes. Gymnastics is a risk factor for increasing EDs. Our results should be taken into consideration by physicians, mental health professionals, sports nutritionists, coaches, and sport medicine specialists. We recommend establishing strategies pertaining to mental health, especially EDs in sports centers, along with screening programs for those who demand additional assessment and supervision.

## 1. Introduction

Eating disorders (EDs) are stated as mental health conditions, referring to an unhealthy relationship between food and weight [1] and a pathogenic and abnormal disturbance in eating behaviors that alter food intake, which inversely affect mental or physical health [2]. These behaviors vary from rigid dieting to full ED syndrome [3,4]. Having serious and harmful physical and psychological consequences puts the spotlight on this pandemic public health threat [5]. Despite having some similarities, disordered eating and eating disorders must be distinguished from one another. Disordered eating describes improper eating behaviors that do not match the criteria for an eating disorder diagnosis, whereas an eating disorder is a clinical diagnostic. Disordered eating behaviors can be seen in persons who have eating disorders; however, not everyone who exhibits these behaviors will have an eating problem [4,5,6].

Athletes are also vulnerably affected by EDs [6]. The type of exercise in some sports types is categorized by weight scale [4,7]. Athletes may participate in lean and non-lean sports; sports focusing on maintaining and accomplishing lower body weight in order to influence athletic performance are known as lean sports, including dancing, swimming, running, and gymnastics, whilst in non-lean sports, there is no need for athletes to be classified as low-weight to be competitive, such as table tennis, horse riding, and basketball [7,8]. Consequently, the leanest athletes are at a higher risk of developing EDs due to uncontrolled pathogenic behaviors to achieve the lowest body weight [7,9]. Generally, the prevalence of EDs doubled in 2006 from 3.5% to 7.8% in 2018 [9], with a massive increase in EDs among female athletes, up to 45%, and male athletes, with 19% in competitive sports [8,9,10]. Recent research suggests that the prevalence of clinically eating pathology was 5.5% amongst elite adolescent athletes [11].

EDs are currently prevalent and are considered prospective problems [12]. It is now crucial to understand the etiology of EDs, which are associated with negative and undesirable consequences, such as high mortality and healthcare costs [13]. The Eating Attitudes Test (EAT) is one of the extensively and commonly self-reported screening tools used to measure disordered eating attitudes and behavior [14,15,16]. This test was shortened and scaled down to 26 items as it was initially established as 40 items to detect anorexia nervosa [16,17]. Shortly afterward, it was validated to be used for bulimia nervosa diagnosing and other types of EDs among the whole population [17,18]. Previously, the Eating Attitudes Test-26 items (EAT-26) showed good specificity and moderate sensitivity to detect and identify the risk of an ED [19]. The EAT-26 includes a three-factor structure, which is the “Dieting” factor, the “Bulimia and food preoccupation” factor, and the “Oral control” factor [18]. EAT-26 has been broadly used in the Middle East. In Arabic countries, the three factors were also found in the recent replication of the sample using Exploratory Structural Equation Models (ESAM) in Jordan, reporting a different pattern in item 0 factor loadings [20]. Indeed, detecting bulimia nervosa, eating disorders that do not binge, or otherwise specified EDs were better at a score of 11 with good sensitivity and specificity [21]. Recently, a cut-off score of 11 is a reasonable withdrawal from the cut-off score of 20, which was previously thought to be indicative of disordered eating tendencies [18].

Despite the huge number of studies recently and previously conducted regarding EDs in western countries [22,23,24] and showing the continuous increase in EDs prevalence, mainly among athletes, none of those were conducted in the Middle East. Accordingly, our study is not just the first but also the only and the most recent research among Arab countries, mainly in Jordan, that spotlighted the ED problem. The study’s purpose was to investigate and identify the EDs prevalence among Jordanian athletes. To the best of the authors’ knowledge, there is no literature or published studies concerning the prevalence of EDs among Jordanian athletes, which awards novelty to our study to be the first regarding Jordan athletes. We hypothesized a hidden incidence of EDs among Jordanian athletes.

## 2. Materials and Methods

The study followed the STROBE-SIIS (Sports Injury and Illness Surveillance) Statement [21]. The International Olympic Committee consensus statement: methods for recording and reporting of epidemiological data on injury and illness in sports 2020 (including the STROBE extension for sports injury and illness surveillance (STROBE-SIIS)) [21]. A cross-sectional, observational study protocol was applied in this research to examine the EAT-26 items tool measurements among adult Jordanian athletes. The study tool was a self-administered, valid, Arabic version, translated questionnaire answered by a convenient participant sample from Jordan (n = 249). The diagnostic questionnaire in this study was the Eating Attitudes Test-26 (EAT-26). EAT-26 is the most popular instrument for assessing the likelihood of developing eating disorders, which was created in 1982 by Garner, Olmsted, Bohr, and Garfinkel. The EAT-26 is an improved version of the EAT-40 exam [17]. For both adolescents and adults, the test shows a high level of validity and test-retest reliability. We used the standardized and validated Arabic translation of the EAT-26 [25].

The aim of our study was to determine the prevalence of “at-risk” EDs among athletes in Jordan. It is well known that unbiased random sampling results are more reliable and provide unbiased conclusions. However, the use of convenience sampling is appropriate for the current research problem as we recruited participants using social media crowdsourcing. Because we had a preset selection criterion (i.e., inclusion and exclusion criteria), applying random sampling would lead to a very small sample size. Sample size calculations indicated that a sample of around 250 participants was sufficient to make powered conclusions. A sample size of approximately 250 is required to guarantee that the 95% confidence interval estimate of the proportion of athletes with an ED is within 5% of the true proportion. Our assumption of the sample size calculation was that 20–25% of the participants would show for an ED based on a recent meta-analysis of university students (used as a proxy for active young adults) [23].

The working definition of an athlete is a person who is professionally engaged in sports and other physical activities. The inclusion criteria were adult athletes over 18 years old, able to read and write in Arabic, provided informative and correct information, and have been engaging in athletic activities for the past two years or more (≥24 months). In addition, they perform exercises 5 days a week and 4 h per day. They are enrolled in sport exercise sessions and not only while being physically active but also all of the participants were either college students or job employees.

The exclusion criteria included any person who did not meet the previous conditions. Participation was fully voluntary, and they were informed that they were free to leave the study.

The following variables were gathered during data collection: Weight and height were self-addressed in this study. Evidence from systematic reviews and meta-analyses, as well as original studies, suggest that the confidence in the accuracy of self-reporting anthropometry data has a strong correlation with clinically measured [26,27]. The body mass index (BMI) (kg/m^2^) was calculated and classified according to the World Health Organization (WHO) categories of underweight, normal, overweight, or obese [28]. Basal energy requirements for all subjects were calculated by the Harris-Benedict equation [29]. BMR (kcal/day) = −780.806 + (11.108 × weight in kg) + (7.164 × height in cm) [30].

Their ideal body weight was calculated by the Robinson Formula [31,32].

Men: Ideal Body Weight (kg) = 52 kg + 1.9 kg per inch over 5 feet.

Women: Ideal Body Weight (kg) = 49 kg + 1.7 kg per inch over 5 feet [33].

Body surface area (BSA) values are commonly used in public health research [34].

BSA (m^2^) = Square root ((Ht (cm) × Wt (kg))/3600).

In this study, it was calculated by the R statistical coding program. The following specific formulas were used:

For the Eating Attitudes Test-26, a cut-off of 20 is used to diagnose cases at risk of an ED based on attitudes, feelings, and behaviors associated with eating. The EAT-26 is a valid tool to measure an ED in early ages, adults, and athletes as special at-risk samples [35,36]. The scale contains three subscales: dieting, bulimia, and food obsession/oral control. The scale scoring values were performed on a 6-point scale from never (0) to always (6). The range sum of EAT-26 scores is from 0 to 78. Scoring of 20 or above was classified as a high risk. Compared to a psychiatric interview, 88% sensitivity and 96% specificity were obtained with the EAT-26 [22].

Data was collected via an online questionnaire through immediate digital conversation using Google forms. The questionnaire used to be in the main despatcher via commercials on social media and instant messaging apps. Other recruitment techniques have been used, including phrase of mouth, work relationships, and authors’ connections. Moreover, members had been requested to share the hyperlink to the questionnaire with extra individuals who match the inclusion standards by using social media. All information had been saved in an impenetrable Google Drive, on hand solely to the researchers, and been coded. The targeted population was the athletes only from different sports types and those who live in Jordan; hence, the results were more representative. The consent form of the online questionnaire listed the inclusion and exclusion criteria. Hence, only the applicable athletes who fit the inclusion criteria answered the questionnaire.

### Statistical Analysis

The data were analyzed by a statistical coding program (R version 4.1.3 (One Push-Up) was released on 10 March 2022), which is a programming language for statistical computing and graphics supported by the R Core Team and the R Foundation for Statistical Computing. Descriptive statistics, including mean and standard deviation or count and proportion, were used to describe the data. Logistic regression analysis was used to identify the factors associated with at-risk eating disorders. Results of the logistic regression are presented as odds ratios and corresponding 95% confidence intervals. A *p*-value < 0.05 was set as statistically significant.

## 3. Results

The EAT-26 questionnaire was completed by 249 athlete participants in September 2022. Enrolment was upon reachable. Participants’ descriptive characteristics are shown in Table 1. Fifty-nine percent were males, and 49% had a normal BMI. The participants were from different parts of Jordan, and we made sure to let our sample represent the whole sports population in Jordan. Thirty-four percent were at risk, with an EAT-26 score of around 18. Within the at-risk group, 18% were over 30 years old, 20% were normal weight, and 10% played gymnastics.

Age categories showed a close risk effect, in which 52% of at-risk participants were over 30 years old. Normal-weight participants, categorized by the BMI classification, represent 58% of the at-risk group; 29% were overweight, 8% were obese, and the least affected were underweight (2%). The ideal body weight for females was calculated to be 95% CI = 56 kg [54, 57 kg]. Interestingly, the highest recorded weight of participants was 95% CI = 79 kg [74, 81 kg]. On the other side, the male’s ideal body weight was 95% CI = 71 kg [70, 72 kg], while their highest weight recorded was 95% CI = 80 kg [77, 83 kg].

Among both sexes, the mean EAT-26 score was close, although females showed a higher value for the upper 95% CI, as shown in Table 1.

According to Table 2, which describes the prevalence of at-risk EDs, 34% of the participants were at-risk of EDs. Of them, 18% out of 34% of total athlete participants (53% of the at-risk participants) were over 30 years old. None of the underweight BMI categories showed to be at risk of EDs. On the other hand, 20% out of 34% of the total participants (59% of the at-risk participants) were at a normal weight, and 10% out of 34% of the total participants (30% of the at-risk athlete participants) were overweight. Hence, the EDs risk increased with the increase in BMI.

The prevalence rate of being at risk of an ED by various sports types is shown in Table 2 and Figure 1. Gymnastics athletes were the most vulnerable sport type to be at risk of EDs, with 10% out of the 34% of total participants (30% of the at-risk athlete participants). In descending order, taekwondo then, mixed sports (multi-type of sports) and swimming are shown in Table 2 and Figure 1. Interestingly, the least prone to be at risk of EDs were the outdoor athletes (outdoor running, outdoor cycling, and outdoor walking), as the percentage was 3% out of 34% of total participants (8% of the at-risk athlete participants). The total number of participants in this study was 249; among them, only 84 were at risk of EDs. Hence in Table 2, all the information listed was among “at-risk” athletes.

According to Table 3, results from logistic regression analyses showed that no variable (i.e., age, sex, current weight, max weight, BMI, or sports years) predicts ED status (all *p* > 0.05). Table 3 represents the “within-the-whole participants differences” with their demographic characteristics (age, sex, and BMI). It was found that all the demographic data did not differ significantly among the 249 participants, which reflects the homogeneity of the sample.

## 4. Discussion

Eating disorders’ expansion raises an alarming, warning, and prevalent issue among different levels of the community and noticeably among athletes [9,37,38]. Fortunately, in the current study, which is the only and the newest study [8,12] among national countries and particularly in Jordan, we aimed to evaluate and estimate the EDs prevalence rate among Jordanian athletes in adulthood period.

Our results addressed a stunning expansion of the EDs among Jordanian athletes. Amongst the responding population, 34% were at risk of EDs. A score at or above 20 on the EAT-26 indicates an “at risk”, and below 20 is considered “low risk”. Scoring 20 or above was classified as a high risk. Our results matched the French data, which demonstrated that 33% of athletes were at risk of EDs [39] and close to Saudi Athletes (i.e., 36.6%) [40]. Finish athletes addressed 18% [41], close to Turkey’s athletes at 19% [42], in Germany at 16.3% [43], and in Australia at 23% [44], while in Greece, the prevalence was the lowest, 5.1% [45]. However, in Norway, the prevalence was 13.5% among the overall athletes’ population, although it was very high among aesthetic sports (i.e., 42%) and lower in endurance sports (i.e., 24%), while the lowest was among athletes who played technical (i.e., 17%) and ball game sports (i.e., 16%) [37]. In the UK [8] and the USA [46], their athletes’ populations on the prevalence of EDs were identical.

We found that the highest prevalence was among gymnastics, with a value of 10% out of 34% of total at-risk participants. This could be explained by their interest and sensitivity toward reducing body weight and thinness, accepting the hypothesis that lean sports competitors are at a higher risk of having an ED and engaging in disordered behaviors to achieve their desired weight [7]. Additionally, the high impact, effort, and pressure put on competitive athletes toward having low body weight and optimizing athletic performance could explain our findings [44]. Consistently to our results, aesthetic sports, such as gymnastics, put a high demand toward being thin and thus are identified as lean sports and correlated to a higher risk of developing an ED; indeed, in these types of sports, athletic performance is evaluated by judges of the competition and appearance of the gymnast is an influencing factor for judging [7]. Moreover, in a scoping review, Karrer et al. (2020) indicated that male elite athletes participating in lean sports had a higher desire toward wanting to be thin, higher levels of restricted diets, and the prevalence of disordered eating was at higher rates compared with non-athletes. Additionally, the prevalence of ED behaviors was elevated in elite male athletes competing in weight-sensitive sports, regarding different types of sports; thus, weight sensitivity leads to uncontrolled eating behaviors and is supposed to be ED’s risk factor [9,47]. Furthermore, Kong and Harris (2015) assumed that in aesthetic sports, such as gymnastics, 60.9% of the athletes were lean athletes and figured out a high prevalence of EDs among those lean athletes [44]. Jordanian athletes showed no differences between the sexes. To optimize athletic performance, lower body weights and lower body fat percentage lead elite athletes to diet rigidly, making them highly vulnerable and exposed to being driven into disordered eating [47]. This would explain the craving for strict diets and disordered behaviors. On the other hand, we found that taekwondo (8% out of 34%) and outdoor sports (3% out of 34%), especially running, were the lowest prevalence. Several studies found a significantly elevated prevalence of EDs among weight-sensitive sports, including wrestling, karate, and judo [46,48,49,50], compared to ball sports and endurance (outdoor such as long-distance running) [8]. Weight-sensitive sports are characterized by dividing competitors into weight categories, and the purpose of this classification is to equalize the differences in strength and agility between competitors [46,50]. The athlete often competes in the weight category that is less than his normal body mass, as it is believed that this will provide them with a competitive advantage over their opponents. Some combat athletes are known to use rapid weight loss strategies before the competition [39,44,50].

The EAT-26 tool can be used in all settings, specifically to assess EDs risk [14]. It can be used for individual assessments or in a group setting and is designed to be managed by psychological health professionals, coaches, athletes, and others interested in gathering information to determine whether an individual is eligible for evaluation and should be referred for EDs [51]. It is ideal for schools, athletic programs, fitness centers, infertility clinics, pediatric offices, general practices, and outpatient mental health departments [18]. It is mainly aimed at young people and adults [52].

It adds new findings and valuable knowledge for Arab athletes and coaches to take into consideration and pay more attention to athletes’ food intake habits and attitudes. It is widely known that nutrition is crucial for these vulnerable groups; hence, our focus was on Jordanian athletes to prevent the undesired, hazardous, and unhealthy outcomes of EDs.

The EAT-26 tool contains 26 items, which is too long to be answered and not applicable from our point of view. Long surveys are pretty common but have a negative effect on response rate and abandonment rate, and it impacts sample representativeness and data quality. Our study would suggest developing a short version of this tool’s item to make it easier for the participants. The short period of collecting the data might be one of the limitations of this study. It is recommended to achieve a larger sample. In addition to the use of social media platforms, we used other recruitment techniques, including phrase of mouth, work relationships, and authors’ connections. Although this variable, i.e., source of recruitment, was not collected during this study we suggest that future studies analyze results from different recruitment strategies.

Strength: This study is the first to assess the risk of eating disorders among Jordanian athletes in Amman, Jordan. It uses the EAT-26 tool, which is one of the most accurate tools in assessing the ED’s risk occurrence.

## 5. Conclusions

The results of the EAT-26 revealed that the prevalence was 34% of athletes were at risk of eating disorders. Age, sex, and BMI did not significantly affect the rates of eating disorders. Outdoor sports had the least effect on Eds, while the highest was amongst gymnastics. We conclude that the EDs prevalence is alarming among Jordanian athletes and should be considered as a mental health problem that would affect the athlete’s performance.

## Figures and Tables

**Figure 1 sports-10-00182-f001:**
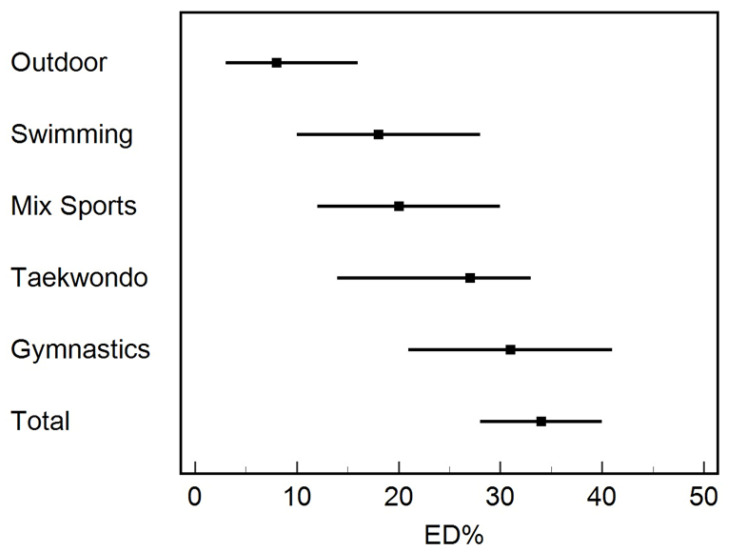
Forest plot of ED among athletes by sports type. Data in Figure 1, corresponding point = point estimate and line = 95% confidence intervals.

**Table 1 sports-10-00182-t001:** Participants’ descriptive data: n = 249.

Female		41%		
Male		59%		
Underweight		4%		
Normal weight		49%		
Overweight		36%		
Obese		12%		
At risk (EAT-26 > 20)		34%		
No risk (EAT-26 < 20)		66%		
			**95% CI**	
	**Mean**	**SD**	**Lower**	**Upper**
Age (years) both sexes	31	8	31	32
Female	32	8	31	34
Male	31	8	30	32
Sports (years) both sexes	8	4	8	9
Female	8	4	8	9
Male	8	4	8	9
Weight (kg) both sexes	74	17	72	76
Female	62	11	60	65
Male	81	16	79	84
Ideal weight * (kg) both sexes	65	10	64	66
Female	56	6	55	57
Male	71	6	70	72
Height (cm) both sexes	171	9	170	172
Female	164	6	163	165
Male	176	7	175	177
BMI (kg/cm ^2^) both sexes	25	4	24	26
Female	23	4	22	24
Male	26	4	26	27
Body surface area ** both sexes	1.9	0.3	1.8	1.9
Female	1.7	0.2	1.7	1.7
Male	2.0	0.2	2.0	2.0
Basal metabolic rate *** (kcal/day) both sexes	1657	267	1624	1691
Female	1478	152	1448	1508
Male	1780	259	1737	1822
Highest weight (kg)	80	16	78	82
Female	78	17	75	82
Male	81	16	78	83
EAT-26 score (20 at risk)	18	10	17	19
Female	18	9	17	20
Male	18	10	16	20

***** Ideal weight was calculated by the Robinson Formula. ****** Body surface area was calculated by the Mosteller Formula. ******* Basal metabolic rate was calculated using the Revised Harris-Benedict Formula.

**Table 2 sports-10-00182-t002:** Prevalence of the EDs among study participants.

Risk	Counts (Out of 249)	% of total
At Risk	84	34%
No Risk	165	66%
**Subgroup Analysis of the Prevalence among At-Risk Participants N = 84 with 34%**		
	**Out of 84**	**Out of 34%**
By age category		
>=30 Years	45	18%
<30 Years	39	16%
By BMI category		
Underweight	1	0%
Normal weight	49	20%
Overweight	26	10%
Obese	8	3%
By sport type		
Outdoor	7	3%
Swimming	15	6%
Mix (multi-type of sports)	17	7%
Taekwondo	19	8%
Gymnastics	26	10%

**Table 3 sports-10-00182-t003:** The association between ED and selected demographic factors.

Variable	OR (95% CI)	*p*-Value
Age	1.02 (0.98, 1.05)	0.38
Sex	1.63 (0.49, 5.44)	0.42
Weight	1.06 (0.89, 1.27)	0.50
Maximum weight	0.95 (0.8, 1.13)	0.57
BMI	0.86 (0.51, 1.44)	0.56
Sports (years)	1.01 (0.95, 1.08)	0.73

ED = defined as EAT-26 > 20.

## Data Availability

Data can be obtained upon request from the first author.

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
