# Peer review of "The Prevalence of “at Risk” Eating Disorders among Athletes in Jordan"

_sports, 2022, doi:10.3390/sports10110182_

Round 1

Reviewer 1 Report

The study’s purpose was to investigate and identify the EDs prevalence among Jordanian athletes and the authors hypothesized a hidden incidence of EDs among Jordanian athletes. Is it possible to say about EDs in the study of 250 random subjects? Why random, I will explain in the next part of the review. 

The inclusion criteria were athletes over 18 years old, able to read and write in Arabic and provide informative, correct information, and has been engaging in athletic activities for the past two years or more. What it means: has been engaging in athletic activities? They train one, two, or seven times a week, have a trainer, the training takes 1-4 hours, and they only train, work, and learn in school.

The body mass index (BMI) (kg/m2) was calculated. Who and how was measured body high and mass?

Other recruitment techniques have been used, including phrase of mouth, work relationships, and authors' connections. Moreover, members had been requested to share the hyperlink to the questionnaire with extra individuals who match the inclusion standards by using social media. Is it a good idea to analyze results from different, random people?

“Table 2. prevalence of the EDs among different characteristics.” Letters should be correct in every place of the manuscript. 

“Descriptive statistics including mean and standard deviation or count, and proportion were used to describe the data. Logistic regression analysis was used to identify the factors associated with at-risk eating disorders. Results of the logistic regression are presented as odds ratio and corresponding 95% confidence intervals. A p-value of < 0.05 was set as statistically significant.” I can’t see the logistic regression results. 

In the results, I see: “According to Table 3, no variable (i.e., age, sex, current weight, max weight, BMI, or sport years) predicts ED status (all p > 0.05)”

BUT in conclusion:

“Age and gender did 250 not significantly affect the rates of eating disorders, while BMI did have a significant impact”.  What is true?

“We found that the highest prevalence was among gymnastics with a value of 10 % out of 34% of total at-risk participants”. Did use any statistical analyses to say it? Are there any statistical differences? 

Author Response

Response to Reviewer 1 Comments

We would like to thank the reviewer for his or her comments. We have addressed each and every comment raised, highlighting changes in RED font for ease of tracking. Also, we mentioned the corresponding line numbers to highlight changes in the response to comments below.

Point 1. The study’s purpose was to investigate and identify the EDs prevalence among Jordanian athletes and the authors hypothesized a hidden incidence of EDs among Jordanian athletes. Is it possible to say about EDs in the study of 250 random subjects? Why random, I will explain in the next part of the review. 

Response 1: We added the following clarification: “The aim of our study was to determine the prevalence of “at risk” eating disorders among athletes in Jordan. It’s well known that unbiased random sampling results in more reliable and unbiased conclusions. However, the use of a convenience sampling is appropriate for the current research problem as we recruited participants using social media crowdsourcing. Because, we had a preset selection criteria (i.e., inclusion and exclusion criteria) applying random sampling will lead to very small sample size. Sample size calculations indicated that a sample of around 250 participants are sufficient to make powered conclusions”. See lines (101-108).

Point 2. The inclusion criteria were athletes over 18 years old, able to read and write in Arabic and provide informative, correct information, and has been engaging in athletic activities for the past two years or more. What it means: has been engaging in athletic activities? They train one, two, or seven times a week, have a trainer, the training takes 1-4 hours, and they only train, work, and learn in school.

Response 2: has been engaged in athletic activities means they are considered athletes and not only physically active. Which means 5 days a week for 4 hours per day. They are adults who train and either work or learn in college. More information has been added to the sampling section line 102-106

Point 3. The body mass index (BMI) (kg/m2) was calculated. Who and how was measured body high and mass?

Response 3: We explained the following: “The following variables were gathered during data collection: Weight and height were self–addressed in this study. Evidence from systematic reviews and meta-analyses as well as original studies suggest that the confidence in accuracy of self-reporting anthropometry data has strong correlation with clinically measured (29,30).”

Point 4. Other recruitment techniques have been used, including phrase of mouth, work relationships, and authors' connections. Moreover, members had been requested to share the hyperlink to the questionnaire with extra individuals who match the inclusion standards by using social media. Is it a good idea to analyze results from different, random people?

Response 4: Thisn points is not possible to analyze as such variable was collected i.e., source of recruitment.  Thus, we added the following statement to limitations: “In addition to the use of social media platforms we used other recruitment techniques have been used, including phrase of mouth, work relationships, and authors' connections. Although this variable i.e., source of recruitment was not collected during this study we suggest that future studies analyze results from different recruitment strategies.” See lines 148,149

Point 5. “Table 2. prevalence of the EDs among different characteristics.” Letters should be correct in every place of the manuscript. 

Response 5: Thank you for this comment. Corrections have been done accordingly; and all of the abbreviations of EDs were corrected.

Point 6. “Descriptive statistics including mean and standard deviation or count, and proportion were used to describe the data. Logistic regression analysis was used to identify the factors associated with at-risk eating disorders. Results of the logistic regression are presented as odds ratio and corresponding 95% confidence intervals. A p-value of < 0.05 was set as statistically significant.” I can’t see the logistic regression results. 

Response 6: Results of the logistic regression are presented in Table 3. We clarified this in line 208.

Point 7. In the results, I see: “According to Table 3, no variable (i.e., age, sex, current weight, max weight, BMI, or sport years) predicts ED status (all p > 0.05)” BUT in conclusion: “Age and gender did 250 not significantly affect the rates of eating disorders, while BMI did have a significant impact”.  What is true?

Response 7: Thank you we removed the confusing statement about BMI. We added the following to the conclusion: “Age, sex, and BMI did not significantly affect the rates of eating disorders.” See line 294.

Point 8. “We found that the highest prevalence was among gymnastics with a value of 10 % out of 34% of total at-risk participants”. Did use any statistical analyses to say it? Are there any statistical differences? 

Response 8: We only used descriptive statistics to describe the trend. No formal testing was performed.  

Reviewer 2 Report

Thank you very much for the opportunity to rate your article.
I consider the subject very interesting, and the results obtained indicate the development of new guidelines for athletes after the end of their sports careers.
However, I indicate two issues that must be resolved in order for the article to be admitted to the next stages of the editorial procedure. 1. The question of motor potential was omitted from the study.
Refer to work in the discussion: Witkowski K., Superson M., Piepiora P. Body composition and motor potential of judo athletes in selected weight categories. Archives of Budo 2021; 17: 161-175. 2. The title requires simplification. You can formulate the title more precisely in a shorter form.

Author Response

Response to Reviewer 2 Comments

Thank you very much for the opportunity to rate your article. I consider the subject very interesting, and the results obtained indicate the development of new guidelines for athletes after the end of their sports careers. However, I indicate two issues that must be resolved in order for the article to be admitted to the next stages of the editorial procedure. The question of motor potential was omitted from the study.

We would like to thank the reviewer for his or her comments. We have addressed each and every comment raised, highlighting changes in RED font for ease of tracking. Also, we mentioned the corresponding line numbers to highlight changes in the response to comments below.

Our team would like to present our appreciation in reviewing our work. We are really glad for your value feedback.

Point 1. Refer to work in the discussion: Witkowski K., Superson M., Piepiora P. Body composition and motor potential of judo athletes in selected weight categories. Archives of Budo 2021; 17: 161-175. 2.

Response 1: reference has been added in line 259 in the discussion section and was listed in the introduction section.

  1. Type of exercise in some sports types are categorized by weight scale (4,8) line 46
  2. Athletes may participate in lean and non-lean sports; sports focusing on maintaining and accomplishing lower body weight in order to influence athletic performance are known as lean sports include dancing, swimming, running and gymnastics whilst in non- lean sports no need for athletes to be classified as low weight to be competitive such as table tennis, horse riding and basketball in line (47-51)
  3. Several studies found a significantly elevated prevalence of EDs among weight-sensitive sports including wrestling, karate, and judo ((4),54,55) compared to ball sports and endurance (outdoor such as long-distance running (8). Line 259

Point 3. The title requires simplification. You can formulate the title more precisely in a shorter form

Response 3: a suggestion title would be (The prevalence of “at risk” Eating Disorders among Athletes in Jordan)

Reviewer 3 Report

This study determined eating disorder risk in Jordanian athletes. There are several major limitations to this study that require revision or clarification:

1)      The EAT-26 measures concerns and symptoms of eating disorders, but cannot be used to diagnose eating disorders. It is more appropriate to state that those with a score of 20 or greater are at risk of eating disorder, but you cannot state that they actually have an eating disorder. The wording on this needs to be changed throughout the manuscript.

2)      The English language requires extensive revision.

Specific comments:

Line 25: Change “EDS” to “EDs”

Line 26: Should “gender” be changed to “sex” here? “Gender” is behavioral whereas “sex” is biological.

Line 26: “Gender and age had no significant differences in the rates of EDs while normal to overweight BMI had a significant effect.” Please clarify the relation between BMI and EDs here (do eating disorders in your study increase or decrease with increasing BMI?).

Line 27: change “lease” to “least”

Line 29: change “wit” to “with”

Line 30: change “nutritionist” to “nutritionists”

Line 38: change “alter” to “altered”

Lines 41-42: Please consider re-writing the sentence “The type of exercise they enrolled in might be divided upon the weight category demanded”. It is unclear what is being said here.

Line 43: “Lean sports consider the weight classes which are believed to affect the athletes’ performance such as dancing, swimming…” The way this sentence is worded, it is implied that these sports have weight classes. Sports such as dancing, swimming, and gymnastics do not have weight classes. I suggest re-wording this sentence to say that body weight control is important in these sports.

Line 45: Here is it stated that body weight is not important in the sport of horse riding. I am not sure this is accurate as racing jockeys are usually more successful if they are small.

Line 49: Should “19” be changed to “19%” here?

Line 51: it is unclear what is meant by “privileged” here.

Line 52: “With the continuous expansion worldwide of prevalent prospective problems…” This is a very broad statement. I suggest re-wording.

Line 55: Change “screening tool” to “screening tools”

Line 58: delete the word “tested”

I have provided suggestions for English language improvement up to this point, but I suggest the authors recruit a colleague who is more proficient in English to edit the remainder of the manuscript.

Lines 62-63: “the three factors were also found in the recent replication in Jordan” What are the “three factors”? It is unclear what is meant by “replication” in this sentence.

Line 84: “The study tool was a self – administered valid Arabic version translated questionnaire…” Is there a reference you could cite to support the validation of the Arabic version of this questionnaire?

Line 94: Please indicate whether the study was approved by an ethics committee and whether participants signed a consent form before participating.

Lines 99 and 101: “In this study, it was calculated by the R-code software language” – it is unclear what is meant here.

Line 109: Change “m2” to “m2” (superscript the “2”). Also, please provide a reference for this equation.

Line 110: “Eating Attitudes Test-26, a cutoff of 20 is used to diagnose ED cases” – As mentioned above, the EAT-26 simply measures risk of eating disorders, but cannot be used to diagnose eating disordes.

Line 142: “Enrolment was upon reachable” – it unclear what is meant by this sentence.

Line 155: Should “genders” be “sexes” here?

In table 1, “at risk” is classified as a score of >20 and “low risk” is classified as a score of <20. What risk category would an athlete be in if they scored 20?

“Total energy requirement” is presented in table 1. Should this be “Basal energy requirement”? Total energy requirement would also depend on energy expended through activity.

Table 2, the column heading states “Prevalence among At-risk participants N=84 with 34% out of 84” Should this be “34% out of 249”?

Line 164: “None of the underweight BMI categories showed to be at risk of EDs.” This is not true, as the number presented in the table is 1.

Figure 1: Please indicate what the error bars in the figure represent

Line 242: “The length of the EAT-26 tool challenged the compatibility of the participants.” – the meaning of this sentence is unclear. This needs to be re-worded.

Line 250: Again, should “gender” be replaced by “sex” here?

Abstract and line 251: Here is it stated BMI had a significant impact on the score; however, Table 3 shows this as not being statistically significant.

Author Response

Response to Reviewer 3 Comments

This study determined eating disorder risk in Jordanian athletes. There are several major limitations to this study that require revision or clarification:

We would like to thank the reviewer for his or her comments. We have addressed each and every comment raised, highlighting changes in RED font for ease of tracking. Also, we mentioned the corresponding line numbers to highlight changes in the response to comments below.

Please find below the responses of your comments.

Point 1. The EAT-26 measures concerns and symptoms of eating disorders, but cannot be used to diagnose eating disorders. It is more appropriate to state that those with a score of 20 or greater are at risk of eating disorder, but you cannot state that they actually have an eating disorder. The wording on this needs to be changed throughout the manuscript.

Response 1 Thank you very much for this comment we added the following statement.

Line 40-45: “Despite having some similarities, disordered eating and eating disorders must be distinguished from one another. Disordered eating describes improper eating behaviors that do not match the criteria for an eating disorder diagnosis, whereas an eating disorder is a clinical diagnostic. Disordered eating behaviors can be seen in persons who have eating disorders, however not everyone who exhibits these behaviors will have an eating problem (4,5,6).”

Changes from eating disorders to at risk of eating disorders have been done throughout the text.

Point 2. The English language requires extensive revision.

Response 2: We have sought English language proof read from a native English speaker. We will wok with MDPI English editing team to ensure flawless production after acceptance.

Point 3. Line 25: Change “EDS” to “EDs”

Response 3: Changes from “EDS” to “EDs” have been done throughout the text.

Point 4. Line 26: Should “gender” be changed to “sex” here? “Gender” is behavioral whereas “sex” is biological.

Response 4: we changed all the “gender” words in the text to “sex”. Changes have been done.

Point 5. Line 26: “Gender and age had no significant differences in the rates of EDs while normal to overweight BMI had a significant effect.” Please clarify the relation between BMI and EDs here (do eating disorders in your study increase or decrease with increasing BMI?).

Response 5: the risk of EDs increases with increasing BMI. A sentence has been added to clarify the text in line 176. Our findings showed that the higher the BMI the more prone to be diagnosed as risk of EDs.

Line 204-207: Table 3 represents the “within-the-whole participants differences” with their demo-graphic characteristics (age, gender and BMI). It was found that all the demographics data didn’t differ significantly among the 249 participants which reflect the homogeneity of the sample.

Line 194-197: On the other hand, 20% out of 34% of the total participants (59% of the at-risk participants) were at a normal weight, and 10% out of 34% of the total participants (30% of the at-risk athletes’ participants) were overweight. Hence, the EDs risk increased with the increase in the BMI.

Point 6. Line 27: change “lease” to “least”

Response 6: Changes have been done.

Point 7. Line 29: change “wit” to “with”

 Response 7: Changes have been done.

Point 8. Line 30: change “nutritionist” to “nutritionists”

Response 8: Changes have been done.

Point 9. Line 38: change “alter” to “altered”

Response 9: Changes have been done.

Point 10. Lines 41-42: Please consider re-writing the sentence “The type of exercise they enrolled in might be divided upon the weight category demanded”. It is unclear what is being said here.

Response 10: We meant that the style of exercise depends on the sport type. If the sport type includes weight categories, athletes would be more concerned of losing or gaining weight. However, we re-wrote the sentence for more clarification: line 46-47: (Type of exercise in some sports types are categorized by weight scale)

Point 11. Line 43: “Lean sports consider the weight classes which are believed to affect the athletes’ performance such as dancing, swimming…” The way this sentence is worded, it is implied that these sports have weight classes. Sports such as dancing, swimming, and gymnastics do not have weight classes. I suggest re-wording this sentence to say that body weight control is important in these sports.

Response 11: gymnastics and swimming in some competition sets are weight considered. However, we changed the sentence to be clearer.

Line 47-51: (Athletes may participate in lean and non-lean sports; sports focusing on maintaining and accomplishing lower body weight in order to influence athletic performance are known as lean sports include dancing, swimming, running and gymnastics whilst in non- lean sports no need for athletes to be classified as low weight to be competitive such as Table tennis, horse riding and basketball.)

Point 12. Line 45: Here is it stated that body weight is not important in the sport of horse riding. I am not sure this is accurate as racing jockeys are usually more successful if they are small.

Response 12: that is correct but the rule of the sport does not require a weight scale. It has been stated that a low body weight isn’t important to be competitive according to this research Mancine RP, Gusfa DW, Moshrefi A, Kennedy SF. Prevalence of disordered eating in athletes categorized by emphasis on leanness and activity type - A systematic review. Vol. 8, Journal of Eating Disorders. BioMed Central Ltd; 2020.

Point 13. Line 49: Should “19” be changed to “19%” here?

Response 13: we added the (%), changes have been done.

Point 14. Line 51: it is unclear what is meant by “privileged” here.

Response 14: we cleared the meaning by re-write the sentence. Changes have been done from (recent research suggested the clinical EDs prevalence to be around 5.5% amongst adolescent privileged athletes) to

Line 56- 57: (A recent- research suggests that the prevalence of clinically eating pathology was 5.5% amongst elite adolescent athletes Recent research suggested the clinical EDs prevalence to be around 5.5% amongst adolescent privileged athletes (24).) 

Point 15. Line 52: “With the continuous expansion worldwide of prevalent prospective problems…” This is a very broad statement. I suggest re-wording.

Response 15: it has been re-written to (EDs are currently prevalent and are considered prospective problems). Changes have been done.

Point 16. Line 55: Change “screening tool” to “screening tools”

Response 16: Changes have been done.

Point 17. Line 58: delete the word “tested”

Response 17: word deleted.

I have provided suggestions for English language improvement up to this point, but I suggest the authors recruit a colleague who is more proficient in English to edit the remainder of the manuscript.

Manuscript was revised by native English speaker.  

 Point 18. Lines 62-63: “the three factors were also found in the recent replication in Jordan” What are the “three factors”? It is unclear what is meant by “replication” in this sentence.

Response 18: A clarification sentence has been added

Line 67- 69: (The EAT-26 includes a three-factor structure which are The “Dieting” factor, The “Bulimia and food preoccupation” factor, and the “Oral control” factor (12).

Point 19. Line 84: “The study tool was a self – administered valid Arabic version translated questionnaire…” Is there a reference you could cite to support the validation of the Arabic version of this questionnaire?

Response 19: we added the following statement: “We used the standardized and validated Arabic translation of the EAT-26.

The reference for Arabic EAT-26 is Al-Adawi S., Dorvlo A.S.S., Burke D.T., Moosa S., Al-Bahlani S. A survey of anorexia nervosa using the Arabic version of the EAT-26 and “gold standard” interviews among Omani adolescents. Eat Weight Disord. 2002;7:304–311.

Point 20. Line 94: Please indicate whether the study was approved by an ethics committee and whether participants signed a consent form before participating.

Response 20: it was approved and the ID of the agreement is written in the Ethical approval section after conclusion. The participants signed the agreement by clicking the agree bottom in the first part of the questionnaire.  Ethical approval: The study was conducted in accordance with the Declaration of Hel-sinki, and approved by the Institutional Review Board (or Ethics Committee) of the Institutional Review Board at the University of Jordan (IRB at UJ) who evaluated the re-search proposal submitted by Hadeel Ali Ghazzawi from the School of Agriculture, Decision No. (13102022): The IRB at the University of Jordan.

Point 21. Lines 99 and 101: “In this study, it was calculated by the R-code software language” – it is unclear what is meant here.

Response 21: We replaced the above statement with the following: “In this study, it was calculated by the R- statistical coding program. The following specific formulas were used:”.

Point 22. Line 109: Change “m2” to “m2” (superscript the “2”). Also, please provide a reference for this equation.

Response 22: Changes have been done. The references are already mentioned in the text and in the reference list. (reference 31)

Line 125-127:  The body mass index (BMI) (kg/m2) was calculated and classified according to the World Health Organization (WHO) categories of underweight, normal, overweight, or obese (31)

Point 23. Line 110: “Eating Attitudes Test-26, a cutoff of 20 is used to diagnose ED cases” – As mentioned above, the EAT-26 simply measures risk of eating disorders, but cannot be used to diagnose eating disorders.

Response 23: thanks for spotting this note. We made it now clearer by re-writing the sentence to (Eating Attitudes Test-26, a cutoff of 20 is used to diagnose cases at risk of ED cases based on attitudes, feelings, and behaviors associated with eating).

Line 138-140 Eating Attitudes Test-26, a cutoff of 20 is used to diagnose cases at risk of ED cases based on attitudes, feelings, and behaviors associated with eating. EAT-26 is a valid tool to measure the ED in early ages, adults, and athletes as special at-risk samples (38,39).

Point 24. Line 142: “Enrolment was upon reachable” – it unclear what is meant by this sentence.

Response 24: it means that participants were enrolled when we could reach them by the online link. Which translates the convenient sampling protocol.

Line 146- 157: Data was collected via an online questionnaire through immediate digital conversation using Google forms. The questionnaire used to be in the main despatcher via commercials on social media and instant messaging apps. Other recruitment techniques have been used, including phrase of mouth, work relationships, and authors' connections. Moreover, members had been requested to share the hyperlink to the questionnaire with extra individuals who match the inclusion standards by using social media. All information had been saved in an impenetrable Google Drive, on hand solely to the researchers, and been coded. the targeted population was the athletes only from different sports types and live in Jordan hence; the results were more representable. The consent form of the online questionnaire listed the inclusion and exclusion criteria. Hence, only the applicable athletes who fit the inclusion criteria answered the questionnaire.

Point 25. Line 155: Should “genders” be “sexes” here?

Response 25: Changes have been done as mentioned in response 4.

Point 26. In Table 1, “at risk” is classified as a score of >20 and “low risk” is classified as a score of <20. What risk category would an athlete be in if they scored 20?

Response 26: A score at or above 20 on the EAT-26 indicates as an “at risk”, below 20 considered “low risk”. As stated in line 138 and line 224: (Scoring of 20 or above was classified as a high risk)

Point 27. “Total energy requirement” is presented in Table 1. Should this be “Basal energy requirement”? Total energy requirement would also depend on energy expended through activity.

Response 27: “Total energy requirement” was changed to “Basal metabolic rate” in Table 1.

Point 28. Table 2, the column heading states “Prevalence among At-risk participants N=84 with 34% out of 84” Should this be “34% out of 249”?

Response 28: n=249 is the total number of participants in this study, among them only 84 were at risk of EDs. Hence in Table 2 all the information listed were among at risk athletes.  Line 204-206

Point 29. Line 164: “None of the underweight BMI categories showed to be at risk of EDs.” This is not true, as the number presented in the Table is 1.

Response 29: Table 1 shows the participant number while in Table 2 it is addressed that the EDs risk was 0% amongst underweight BMI participants.

Point 30. Figure 1: Please indicate what the error bars in the figure represent

  Response 30: We mentioned that errors correspond to 95% confidence interval. “Data in Figure 1 correspond point = point estimate and line = 95% confidence intervals”.

Point 31. Line 242: “The length of the EAT-26 tool challenged the compatibility of the participants.” – the meaning of this sentence is unclear. This needs to be re-worded.

Response 31: Line 280-283: modified to (The EAT-26 tool contains 26 items which is too long to be answered and not applicable from our point of view. Long surveys are pretty common but have a negative effect on response rate, abandonment rate, impacting sample representativeness, and data quality.

Point 32. Line 250: Again, should “gender” be replaced by “sex” here?

Response 32: Changes have been done as stated in responses 4 and 24.

Point 33. Abstract and line 251: Here is it stated BMI had a significant impact on the score; however, Table 3 shows this as not being statistically significant.

Response 33: We removed this confusing statement about BMI.  

Round 2

Reviewer 1 Report

This version is better but in the title of table 2, the first letter should be changed. In my opinion analyzes so different athletes is very difficult.

"Table 2. prevalence of the EDs among different characteristics"

Author Response

Response to Reviewer 1 Comments

We would like to thank the reviewer for his or her comments.

We have addressed the minor comments raised, highlighting changes in RED font for ease of tracking.

Point-by-point response is below:

This version is better but in the title of table 2, the first letter should be changed. In my opinion analyzes so different athletes is very difficult. "Table 2. prevalence of the EDs among different characteristics"

Thank you for your nice comment. We capitalized the word Prevalence in Table 2.

We also clarified that the remaining of the table is “Subgroup analysis of the prevalence among at-risk participants N=84 with 34%”. The analysis used is basic descriptive to allow clear understanding of the results.

Thank you for your comments and support.

Reviewer 3 Report

Thanks for addressing my previous comments.

In the abstract, I think you still need to clearly state that you are measuring risk of eating disorders and not actually eating disorders in the athletes.

In the conclusion, line 295: 34% of what? You need to add something here...should this say "34% of athletes at risk of eating disorders"?

Line 300: "preys'" is not the correct word to use here. Please correct this. On this line you also indicated that BMI is important to consider, which contradicts line 295.

Author Response

Response to Reviewer 3 Comments

We would like to thank the reviewer for his or her comments.

We have addressed the minor comments raised, highlighting changes in RED font for ease of tracking.

Point-by-point response is below:

Thanks for addressing my previous comments.

Thank you for your nice comment.

In the abstract, I think you still need to clearly state that you are measuring risk of eating disorders and not actually eating disorders in the athletes.

We have added this explanation in the abstract: “While an ED is a clinical psychiatric diagnosis that can be only established after a psychiatric assessment, it is important to note that "at-risk" refers to people who will exhibit aberrant eating patterns but do not fully meet the requirements for an ED diagnosis. This study was designed to address the EDs symptoms (i.e., “at-risk”) in Jordanian athletes, and their association with age, sex, body mass index (BMI), and type of sport.

In the conclusion, line 295: 34% of what? You need to add something here...should this say "34% of athletes at risk of eating disorders"?

We have completed the statement as follow: “The results of the EAT-26 revealed that the prevalence was 34% of athletes at risk of eating disorders."

Line 300: "preys'" is not the correct word to use here. Please correct this. On this line you also indicated that BMI is important to consider, which contradicts line 295.

We have deleted this sentence/statement to avoid the contradiction.